# TRGP: Trust Region Gradient Projection for Continual Learning

**Sen Lin**[1], **Li Yang**[1], **Deliang Fan**[1], **Junshan Zhang**[1,2]
[1]School of ECEE, Arizona State University, [2]Department of ECE, University of California, Davis
{slin70, lyang166, dfan}@asu.edu, jazh@ucdavis.edu

## Abstract

Catastrophic forgetting is one of the major challenges in continual learning. To address this issue, some existing methods put restrictive constraints on the optimization space of the new task for minimizing the interference to old tasks. However, this may lead to unsatisfactory performance for the new task, especially when the new task is strongly correlated with old tasks. To tackle this challenge, we propose Trust Region Gradient Projection (TRGP) for continual learning to facilitate the forward knowledge transfer based on an efficient characterization of task correlation. Particularly, we introduce a notion of 'trust region' to select the most related old tasks for the new task in a layer-wise and single-shot manner, using the norm of gradient projection onto the subspace spanned by task inputs. Then, a scaled weight projection is proposed to cleverly reuse the frozen weights of the selected old tasks in the trust region through a layer-wise scaling matrix. By jointly optimizing the scaling matrices and the model, where the model is updated along the directions orthogonal to the subspaces of old tasks, TRGP can effectively prompt knowledge transfer without forgetting. Extensive experiments show that our approach achieves significant improvement over related state-of-the-art methods.

## 1 Introduction

Human beings can continuously learn different new tasks without forgetting the learnt knowledge of old tasks in their lifespan. Aiming to achieve this remarkable capability for the deep neural networks (DNNs), continual learning (CL) (Chen & Liu, 2018) has garnered much attention in recent years. Nevertheless, many existing CL methods still leave the DNN vulnerable to forget the knowledge of old tasks when learning new tasks. Such a phenomenon is known as 'Catastrophic Forgetting' (McCloskey & Cohen, 1989), which has become one of the major challenges for CL.

Many approaches (e.g., (Rusu et al., 2016; Li & Hoiem, 2017; Dhar et al., 2019; Guo et al., 2020; Zeng et al., 2019)) have been proposed to address the forgetting issue, which can be generally divided into two classes depending on the network architecture, i.e., expansion methods and non-expansion methods. In order to understand the fundamental limit of a fixed capacity neural network, we focus on non-expansion methods in this work. The basic idea for non-expansion methods is to constrain the gradient update either explicitly or implicitly when learning the new task, so as to minimize the introduced interference to old tasks. For example, the regularization-based methods (e.g., (Kirkpatrick et al., 2017; Serra et al., 2018)) penalize the modification on the most important weights of old tasks through model regularizations; experience-replay based methods (e.g., (Shin et al., 2017; Chaudhry et al., 2019)) constrain the gradient directions by replaying the data of old tasks during learning of new tasks, in the format of either real data or synthetic data from generative models; and orthogonal-projection based methods (e.g., (Farajtabar et al., 2020; Saha et al., 2021)) update the model with gradients in the orthogonal directions of old tasks, without the access to old task data. In particular, the recently proposed Gradient Projection Memory (GPM) (Saha et al., 2021) has demonstrated superior performance compared to other approaches.

To sufficiently minimize the interference to old tasks, most existing non-expansion methods (particularly the orthogonal-projection based methods), often put restrictive constraints on the optimization space of the new task, which may throttle the learning performance for the new task. A plausible conjecture is that such a scenario is likely to occur when the new task is strongly correlated with

old tasks, and in this study we provide evidence to support this conjecture. The underlying rationale is as follows: The weights that are important to the new task are also important to the old tasks strongly correlated with the new task, which are often frozen to address the forgetting in the existing methods; however, they should be updated in the learning of the new task.

To tackle this challenge, a key insight is that for a new task that is strongly correlated with old tasks, although the model optimization space could be more restrictive, there should be *better forward knowledge transfer* from the correlated old tasks to the new task. With this insight, we propose an innovate continual learning approach to facilitate the forward knowledge transfer without forgetting. The main contributions can be summarized as follows:

(1) Inspired by (Schulman et al., 2015), we introduce a novel notion of 'trust region' based on the norm of gradient projection onto the subspace spanned by task inputs, which selects the old tasks strongly correlated to the new task in a layer-wise and single-shot manner. Intuitively, the new task and the selected old tasks in the trust region have similar input features for the corresponding layer.

(2) We propose a novel approach for the new task to leverage the knowledge of the strongly correlated old tasks in the trust region through a scaled weight projection. Particularly, a scaling matrix is learnt in each layer for the new task to scale the weight projection onto the subspace of old tasks in the trust region, in order to reuse the frozen weights of old tasks without modifying the model.

(3) Building on the introduced trust region, scaled weight projection, and a module to construct task input subspace, we develop a continual learning approach, trust region gradient projection (TRGP), that jointly optimizes the scaling matrices and the model for the new task. To mitigate the forgetting issue further, the model is updated along the directions orthogonal to the subspaces of old tasks.

(4) We evaluate TRGP on standard CL benchmarks using various network architectures. Compared to related state-of-the-art approaches, TRGP achieves substantial performance improvement on all benchmarks, and demonstrates universal improvement on all tasks. The superior performance indicates that TRGP can effectively promote the forward knowledge transfer while alleviating forgetting.

## 2 RELATED WORK

**Expansion-based methods.** Expansion-based methods (e.g., (Rusu et al., 2016; Li & Hoiem, 2017; Rosenfeld & Tsotsos, 2018; Hung et al., 2019; Yoon et al., 2017; Li et al., 2019; Veniat et al., 2020)) dynamically expand the network capacity to reduce the interference between the new tasks and the old ones. Progressive Neural Network (PNN) (Rusu et al., 2016) expands the network architecture for new tasks and preserves the weights of old tasks. Learning Without Forgetting (LWF) (Li & Hoiem, 2017) splits the model layers into two parts, i.e., the shared part co-used by all tasks, and the task-specific part which grows for new tasks. Dynamic-Expansion Net (DEN) (Yoon et al., 2017) and Compacting-Picking-Growing (CPG) (Hung et al., 2019) combine the strategies of model compression/pruning, weight selection and model expansion. In order to find the optimal structure for each of the sequential tasks, Reinforced Continual Learning (RCL) (Xu & Zhu, 2018) leverages reinforcement learning and (Li et al., 2019) adapts architecture search. APD (Yoon et al., 2020) adds additional task-specific parameters for each task and selectively learns the task-shared parameters.

**Regularization-based methods.** This category of methods (e.g., (Kirkpatrick et al., 2017; Lee et al., 2017; Chaudhry et al., 2018a; Dhar et al., 2019; Ritter et al., 2018; Schwarz et al., 2018; Zenke et al., 2017)) protect the old tasks by adding regularization terms in the loss function to penalize the model change on their important weights. Notably, to determine the weight importance, Elastic Weight Consolidation (EWC) (Kirkpatrick et al., 2017) leverages Fisher information matrix, HAT (Serra et al., 2018) learns hard attention masks. MAS (Aljundi et al., 2018) evaluates the model outputs sensitivity to the inputs in an unsupervised manner.

**Memory-based methods.** Depending on if data of old tasks is utilized when learning new tasks, memory-based methods can be further divided into the following two categories. *1) Experience-replay based methods.* This class of methods replays the old tasks data along with the current task data to mitigate catastrophic forgetting. Gradient Episodic Memory (GEM) (Lopez-Paz & Ranzato, 2017) and Averaged GEM (A-GEM) (Chaudhry et al., 2018b) alter the current gradient based on the gradient computed with data in the memory. A unified view of episodic memory based methods is proposed in (Guo et al., 2020), based on new approaches are developed to balance between old tasks and the new task. Tiny episodic memory is considered in (Chaudhry et al., 2019) and meta-

learning is leveraged in (Riemer et al., 2018). *2) Orthogonal-projection based method.* To eliminate the need of storing data of old tasks, recently a series work (Zeng et al., 2019; Farajtabar et al., 2020; Saha et al., 2021) updates the model in the orthogonal direction of old tasks, and has shown remarkable performance. Particularly, Orthogonal Weight Modulation (OWM) (Zeng et al., 2019) learns a projector matrix to multiply with the new gradients. Orthogonal Gradient Descent (OGD) (Farajtabar et al., 2020) stores the gradient directions of old tasks and projects the new gradients on the directions orthogonal to the subspace spanned by the old gradients. Gradient Projection Memory (GPM) (Saha et al., 2021) stores the bases of the subspaces spanned by old task data and projects the new gradients on the directions orthogonal to these subspaces.

## 3 PROBLEM FORMULATION

**Continual learning.** Consider the setting where a sequence of tasks $\mathbb{T} = \{t\}_{t=1}^T$ arrives sequentially. Each task $t$ has a dataset $\mathbb{D}_t = \{(\boldsymbol{x}_{t,i}, \boldsymbol{y}_{t,i})\}_{i=1}^{N_t}$ with $N_t$ sample pairs, where $\boldsymbol{x}_{t,i}$ is the input vector and $\boldsymbol{y}_{t,i}$ is the label vector. Consider **a fixed capacity** neural network with $L$ layers, and the set of weights is denoted as $\mathbb{W} = \{\boldsymbol{W}^l\}_{l=1}^L$, where $\boldsymbol{W}^l$ is the layer-wise weight for layer $l$. Given the data input $\boldsymbol{x}_{t,i}$ for task $t$, denote $\boldsymbol{x}_{t,i}^l$ as the input of layer $l$ and $\boldsymbol{x}_{t,i}^1 = \boldsymbol{x}_{t,i}$. The output $\boldsymbol{x}_{t,i}^{l+1}$ for layer $l$ is computed as $\boldsymbol{x}_{t,i}^{l+1} = f(\boldsymbol{W}^l, \boldsymbol{x}_{t,i}^l)$, where $f$ is the operation of the network layer. Following (Saha et al., 2021), we denote $\boldsymbol{x}_{t,i}^l$ as the **representations** of $\boldsymbol{x}_{t,i}$ at layer $l$. When learning task $t$, we only have access to dataset $\mathbb{D}_t$. Let $\mathcal{L}(\mathbb{W}, \{(\boldsymbol{x}_{t,i}, \boldsymbol{y}_{t,i})\}) = \mathcal{L}_t(\mathbb{W})$ denote the loss function for training, e.g., mean squared and cross-entropy loss, and $\mathbb{W}_t$ denote the model after learning task $t$.

**Orthogonal-projection based methods.** To minimize the interference to old tasks, recently a series of studies (Zeng et al., 2019; Farajtabar et al., 2020; Saha et al., 2021) has been carried out to update the model for the new task in the direction orthogonal to the subspace spanned by inputs of old tasks. In what follows, we briefly introduce the main ideas through a basic case with two tasks 1 and 2.

Denote the subspace spanned by the inputs of task 1 for layer $l$ as $S_1^l$ and the learnt model for task 1 as $\mathbb{W}_1 = \{\boldsymbol{W}_1^l\}_{l=1}^L$. It is clear that $\boldsymbol{x}_{1,i}^l \in S_1^l$. When learning task 2, the model $\boldsymbol{W}_1^l$ will be modified in the direction orthogonal to $S_1^l$, by either multiplying the gradient $\nabla_{\boldsymbol{W}^l}\mathcal{L}_2$ with a projector matrix (e.g, (Zeng et al., 2019)), or projecting the gradient $\nabla_{\boldsymbol{W}^l}\mathcal{L}_2$ onto the orthogonal direction to $S_1^l$ (e.g., (Saha et al., 2021)). Let $\Delta\boldsymbol{W}_1^l$ denote the model change after learning task 2. It follows immediately that $\Delta\boldsymbol{W}_1^l \boldsymbol{x}_{1,i}^l = 0$, and the model $\boldsymbol{W}_2^l$ for task 2 is $\boldsymbol{W}_2^l = \boldsymbol{W}_1^l + \Delta\boldsymbol{W}_1^l$. Therefore, for task 1:

$$\boldsymbol{W}_2^l \boldsymbol{x}_{1,i}^l = (\boldsymbol{W}_1^l + \Delta\boldsymbol{W}_1^l)\boldsymbol{x}_{1,i}^l = \boldsymbol{W}_1^l \boldsymbol{x}_{1,i}^l + \Delta\boldsymbol{W}_1^l \boldsymbol{x}_{1,i}^l = \boldsymbol{W}_1^l \boldsymbol{x}_{1,i}^l, \tag{1}$$

which indicates that no interference is introduced to task 1 after learning task 2, thereby addressing the forgetting issue. Such an analysis can be generalized to a sequence of tasks.

**When would orthogonal projection hinder the learning of a new task?** Orthogonal projection provides a promising solution to address the forgetting in continual learning. However, by modifying the model only in the orthogonal direction to the input space of old tasks, the optimization space of learning the new task could be more restrictive, resulting in compromised performance of the new task. To get a more concrete sense, consider the following basic examples with two tasks 1 and 2.

*(Toy example 1)* Suppose task 1 has dataset $\mathbb{D}_1 = \{(\boldsymbol{x}_i, \boldsymbol{y}_i)\}_{i=1}^N$ and task 2 has dataset $\mathbb{D}_2 = \{(-\boldsymbol{x}_i, \boldsymbol{y}_i)\}_{i=1}^N$, where only the sign is changed for the input vectors. Consider the case where two tasks share the same classifier (Saha et al., 2021). It is clear that for the $l$-th layer, the subspace spanned by $\{\boldsymbol{x}_i^l\}_{i=1}^N$ of task 1 is same with the subspace spanned by $\{-\boldsymbol{x}_i^l\}_{i=1}^N$ of task 2, i.e., $S_1^l = S_2^l$, given the learnt model $\boldsymbol{W}_1^l$ for task 1. Based on the fact that stochastic gradient descent updates lie in the subspace spanned by the data input (Zhang et al., 2021; Saha et al., 2021), it follows that the gradient $\nabla_{\boldsymbol{W}^l}\mathcal{L}_2 \in S_2^l$, such that $\nabla_{\boldsymbol{W}^l}\mathcal{L}_2 \in S_1^l$. Therefore, the projection of $\nabla_{\boldsymbol{W}^l}\mathcal{L}_2$ onto the orthogonal direction to $S_1^l$ is 0, which means that the model $\boldsymbol{W}_1^l$ will not be updated when learning task 2, i.e., $\boldsymbol{W}_2^l = \boldsymbol{W}_1^l$. However, the optimal model for task 2 should be $\boldsymbol{W}_2^l = -\boldsymbol{W}_1^l$, because $\boldsymbol{W}_1^l \boldsymbol{x}_i^l$ achieves the minimum loss for the label $\boldsymbol{y}_i$ after learning task 1.

*(Toy example 2)* Suppose the input subspace of task 1 is orthogonal to that of task 2, i.e., $S_1^l \perp S_2^l$. It follows that the projection of $\nabla_{\boldsymbol{W}^l}\mathcal{L}_2$ onto the orthogonal direction to $S_1^l$ is indeed equal to $\nabla_{\boldsymbol{W}^l}\mathcal{L}_2$. Consequently, updating the model for task 2 based on orthogonal projection will not only introduce no interference to task 1, but also move along the direction of steepest descent for task 2.

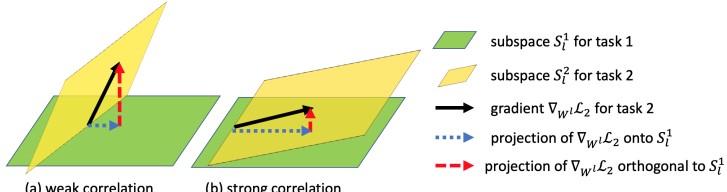

Figure 1: Layer-wise task correlation for the case where the subspace spanned by the representations is a two-dimensional plane. The subspaces are weakly correlated if they are nearly orthogonal and strongly correlated if they are nearly parallel.

Motivated by these examples, a plausible conjecture is that *naive orthogonal projection could possibly compromise the learning performance of the new task that is strongly correlated with old tasks, especially when the correlation is "negative" as in the toy example 1.* In this study, we advocate to characterize the task correlation through the correlation between the input subspaces for two tasks. As illustrated in Fig. 1, when the subspace is 2-dimensional, two tasks are weakly correlated if their input subspaces are nearly orthogonal, and strongly correlated if their subspaces are nearly parallel.

## 4 TRUST REGION GRADIENT PROJECTION FOR CONTINUAL LEARNING

To tackle these challenges, a key insight is that *for a new task that is strongly correlated with old tasks, although the model optimization space could be more restrictive, there should be better forward knowledge transfer from the correlated old tasks to the new task.* With this insight, we propose a novel approach to prompt forward knowledge transfer without forgetting, by 1) introducing a novel notion of trust region to select the most related old tasks in a single-shot manner and 2) cleverly reusing the frozen weights of the selected tasks in the trust region with a scaled weight projection.

### 4.1 TRUST REGION

To facilitate forward knowledge transfer from the correlated old tasks to the new task, the first question is how to efficiently select the most correlated old tasks. Towards this end, we characterize the correlation between the input subspaces for two tasks, through the lens of gradient projection.

Specifically, denote $S_j^l = span\{\boldsymbol{B}_j^l\}$ as the subspace spanned by the task $j$ data for layer $l$, where $\boldsymbol{B}_j^l = [\boldsymbol{u}_{j,1}^l, ..., \boldsymbol{u}_{j,M_{j,l}}^l]$ is the bases for $S_j^l$ (totally $M_{j,l}$ bases extracted from the input). For any matrix $\boldsymbol{A}$ with a suitable dimension, denote its projection onto the subspace $S_j^l$ as:

$$\text{Proj}_{S_j^l}(\boldsymbol{A}) = \boldsymbol{A}\boldsymbol{B}_j^l(\boldsymbol{B}_j^l)' \tag{2}$$

where $(\cdot)'$ is the matrix transpose. We next define a layer-wise trust region for a new task as a set of its most related old tasks, based on the norm of projected gradient onto the subspaces of old tasks.

**Definition 1 (Layer-Wise Trust Region).** *For any new task $t \geq 2$ and layer $l$, we define a layer-wise trust region $\mathcal{TR}_t^l = \{j\}$ for $j \in [1, t-1]$, where for any task $j \in \mathcal{TR}_t^l$ the following holds:*

$$\|\text{Proj}_{S_j^l}(\nabla_{\boldsymbol{W}^l}\mathcal{L}_t(\mathbb{W}_{t-1}))\|_2 \geq \epsilon^l \|\nabla_{\boldsymbol{W}^l}\mathcal{L}_t(\mathbb{W}_{t-1})\|_2, \tag{3}$$

*where $\epsilon^l \in [0, 1]$ and $\mathbb{W}_{t-1}$ is the model after learning task $t-1$.*

Intuitively, for the new task $t$, the norm of its gradient projection onto the subspace of an old task $j$ serves as a surrogate for characterizing the correlation between input subspaces for these two tasks, due to the fact that the gradient lies in the span of its input. When the condition Eq. (3) is satisfied, the gradient $\nabla_{\boldsymbol{W}^l}\mathcal{L}_t(\mathbb{W}_{t-1})$ has a large projection onto the subspace of an old task $j$, which implies that the subspace $S_t^l$ for task $t$ and the subspace $S_j^l$ for task $j$ may have sufficient common bases for layer $l$. In this case, we trust that the old task

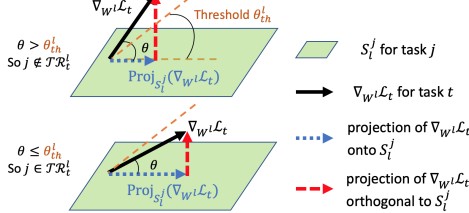

Figure 2: Trust region for a 2-dimensional subspace can be interpreted as: if the angle $\theta$ between $\nabla_{\boldsymbol{W}^l}\mathcal{L}_t$ and $\text{Proj}_{S_j^l}(\nabla_{\boldsymbol{W}^l}\mathcal{L}_t)$ is less than $\theta_{th}^l$ (larger projection on $S_j^l$), old task $j$ is selected to $\mathcal{TR}_t^l$; otherwise not. $\theta_{th}^l$ can be set as a large value, and we can pick tasks with top-$K$ smallest $\theta$ to $\mathcal{TR}_t^l$.

$j$ is strongly correlated with the new task $t$ in layer $l$, and put it into task $t$'s trust region $\mathcal{TR}_t^l$. A simple illustration of trust region is shown in Figure 2. Note that the notion of trust region can also be generalized to a task-wise definition, where the most correlated old tasks will be selected based on the projection of the entire gradient $\nabla_{\mathbb{W}}\mathcal{L}_t(\mathbb{W}_{t-1})$. However, the layer-wise trust region could select different tasks for different layers, which provides a more fine-resolution characterization of task correlations in terms of layer-level features.

**Practical implementation.** Besides the valuable functionality provided by the trust region for selecting most correlated old tasks, another significant benefit is the simplicity of its practical implementation. Consider the implementation for learning a new task $t$.

(1) *Single-shot manner.* Given the learnt model $\mathbb{W}_{t-1}$, we select a sample batch from dataset $\mathbb{D}_t$, and compute the gradient $\nabla_{\boldsymbol{W}^l}\mathcal{L}_t(\mathbb{W}_{t-1})$ in one forward-backward pass for all layers at once. Given the subspace $S_j^l$ for an old task $j$, the condition Eq. (3) can be immediately evaluated for all old tasks.

(2) *Top-K correlated tasks.* It is clear that the choice of $\epsilon^l$ has a nontrivial impact on the selection of the most correlated old tasks. To reduce the sensitivity of the performance on $\epsilon^l$, we can set a relatively small value of $\epsilon^l$, and pick the top-$K$ old tasks with the largest gradient projection norm $\|\text{Proj}_{S_j^l}(\nabla_{\boldsymbol{W}^l}\mathcal{L}_t(\mathbb{W}_{t-1}))\|_2$ from the tasks satisfying Eq. (3). As demonstrated later in our experiments, setting $K = 1$ is enough to achieve a significant performance improvement.

## 4.2 Scaled Weight Projection

Given the layer-wise trust region $\mathcal{TR}_t^l$ for the new task $t$, the next key question is how to efficiently leverage the knowledge of the most correlated old tasks in $\mathcal{TR}_t^l$ for learning task $t$. To this end, we propose a novel approach to reuse the frozen weights of the selected old tasks in $\mathcal{TR}_t^l$ through a scaled weight projection with a scaling matrix.

At the outset, it is of interest to understand what knowledge is preserved for old tasks during continual learning with orthogonal projection. Based on Eq. (1) for the simple case with two learning tasks 1 and 2 as mentioned earlier, it can be shown that

$$\text{Proj}_{S_1^l}(\boldsymbol{W}_2^l) = \text{Proj}_{S_1^l}(\boldsymbol{W}_1^l + \Delta\boldsymbol{W}_1^l) = \text{Proj}_{S_1^l}(\boldsymbol{W}_1^l) + \text{Proj}_{S_1^l}(\Delta\boldsymbol{W}_1^l) = \text{Proj}_{S_1^l}(\boldsymbol{W}_1^l) \qquad (4)$$

where the last equation holds because the model $\boldsymbol{W}_1^l$ is updated in the direction orthogonal to $S_1^l$ when learning task 2. By generalizing Eq. (4) to the case with a sequence of tasks, we can have that for the model $\mathbb{W}_{t-1}$ after learning task $t-1$ and any old task $j < t$:

$$\text{Proj}_{S_j^l}(\boldsymbol{W}_{t-1}^l) = \text{Proj}_{S_j^l}(\boldsymbol{W}_j^l), \qquad (5)$$

which indicates that *the model weight projection on the subspace of old tasks is actually "frozen" during continual learning so as to overcome forgetting of the old tasks.*

On the other hand, because the trust region $\mathcal{TR}_t^l$ is constructed in a way that the subspace $S_t^l$ of task $t$ is strongly correlated with the subspace $S_j^l$ for any old task $j \in \mathcal{TR}_t^l$, the bases $\boldsymbol{B}_j^l$ of $S_j^l$ is very likely to contain important bases for task $t$. As a result, the weight projection $\text{Proj}_{S_j^l}(\boldsymbol{W}_{t-1}^l)$ is *important for the new task $t$* and should be modified accordingly in order to guarantee the learning performance of task $t$, which however has to be frozen to protect task $j$. To find an efficient way to leverage $\text{Proj}_{S_j^l}(\boldsymbol{W}_{t-1}^l)$ without modifying it, note that the projection $\text{Proj}_{S_j^l}(\boldsymbol{W}_{t-1}^l)$ is indeed a linear combination of the projection of $\boldsymbol{W}_{t-1}^l$ onto each basis in $\boldsymbol{B}_j^l$, and every point in $S_j^l$ can be obtained by scaling the coordinates of $\text{Proj}_{S_j^l}(\boldsymbol{W}_{t-1}^l)$. Figure 3 shows a simple example for two-dimensional subspace. Therefore, we propose a scaled weight projection to find the best point for task $t$ in $S_j^l$ by leveraging the projection $\text{Proj}_{S_j^l}(\boldsymbol{W}_{t-1}^l)$ through a square scaling matrix $\boldsymbol{Q}_{j,t}^l$:

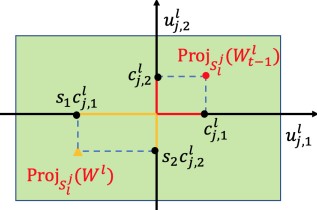

Figure 3: $[\boldsymbol{u}_{j,1}^l, \boldsymbol{u}_{j,2}^l]$ is the bases of subspace $S_j^l$, and $[c_{j,1}^l, c_{j,2}^l]$ is the coordinate of $\text{Proj}_{s_j^l}(\boldsymbol{W}_{t-1}^l)$. Any point $\text{Proj}_{S_j^l}(\boldsymbol{W}^l)$ in $S_j^l$ can be obtained by scaling the coordinate $[c_{j,1}^l, c_{j,2}^l]$ with some scalar $s_1$ and $s_2$.

$$\text{Proj}_{S_j^l}^Q(\boldsymbol{W}_{t-1}^l) = \boldsymbol{W}_{t-1}^l \boldsymbol{B}_j^l \boldsymbol{Q}_{j,t}^l (\boldsymbol{B}_j^l)'. \qquad (6)$$

The dimension of $Q_{j,t}^l$ depends on the number of bases in $B_j^l$ (dimension of $S_j^l$), which is usually small for each task. In this way, we explicitly transfer the knowledge of the selected old tasks in the trust region $\mathcal{TR}_t^l$ to the new task $t$ through a scaling matrix $Q_{j,t}^l$.

## 4.3 TASK SUBSPACE CONSTRUCTION

To successfully leverage the trust region, a missing ingredient is the construction of input subspaces of old tasks. We next show how the subspace $S_j^l$ can be constructed for task $j$ at layer $l$.

**For task $j = 1$.** As in (Saha et al., 2021), we obtain the bases $B_1^l$ after learning task 1 using Singular Value Decomposition (SVD) on the representations. Specifically, given the model $\mathbb{W}_1$ after learning task 1, we construct a representation matrix $R_1^l = [x_{1,1}^l, ..., x_{1,n}^l] \in \mathbb{R}^{m \times n}$ with $n$ samples, where each $x_{1,i}^l \in \mathbb{R}^m$, is the representation at layer $l$ by forwarding the sample $x_{1,i}$ through the network. Then, we apply SVD to the matrix $R_1^l$, i.e., $R_1^l = U_1^l \Sigma_1^l (V_1^l)'$, where $U_1^l = [u_{1,1}^l, ..., u_{1,m}^l] \in \mathbb{R}^{m \times m}$ is an orthogonal matrix with left singular vector $u_{1,i}^l \in \mathbb{R}^m$, $V_1^l = [v_{1,1}^l, ..., v_{1,n}^l] \in \mathbb{R}^{n \times n}$ is an orthogonal matrix with right singular vector $v_{1,i}^l \in \mathbb{R}^n$, and $\Sigma_1^l \in \mathbb{R}^{m \times n}$ is a rectangular diagonal matrix with non-negative singular values $\{\sigma_{1,i}^l\}_{i=1}^{\min\{m,n\}}$ on the diagonal in a descending order. To obtain the bases for subspace $S_1^l$, we use $k_1^l$-rank matrix approximation to pick the first $k_1^l$ left singular vectors in $U_1^l$, such that the following condition is satisfied for a threshold $\eta_{th}^l \in (0, 1)$:

$$\|(R_1^l)_{k_1^l}\|_F^2 \geq \epsilon_{th}^l \|R_1^l\|_F^2 \tag{7}$$

where $(R_1^l)_{k_1^l} = \sum_{i=1}^{k_1^l} \sigma_{1,i} u_{1,i}^l (v_{1,i}^l)'$ is a $k_1^l$-rank ($k_1^l \leq r$) approximation of the representation matrix $R_1^l$ with rank $r \leq \min\{m,n\}$, and $\|\cdot\|_F$ is the Frobenius norm. Then the bases $B_1^l$ for subspace $S_1^l$ can be constructed as $B_1^l = [u_{1,1}^l, ..., u_{1,k_1^l}^l]$.

**For task $j \in [2, T]$.** We construct the bases $B_j^l$ after learning task $j$ given the learnt model $\mathbb{W}_j$. A representation matrix $R_j^l$ will be first obtained in the same manner as $R_1^l$. Note that the bases $\{B_i^l\}_{i=1}^{j-1}$ learnt for old tasks may include important bases for task $j$. Therefore, we learn the bases $B_j^l$ by selecting the most important bases from both bases of old tasks and newly constructed bases. Specifically, (1) (**old bases**) we first concatenate the bases $\{B_i^l\}_{i=1}^{j-1}$ of old tasks together in $M_j^l$ and eliminate the common bases. For each basis $u_i^l \in M_j^l$, we compute the corresponding eigenvalue of $R_j^l(R_j^l)'$, i.e., $\delta_i^l = (u_i^l)' R_j^l (R_j^l)' u_i^l$, which is the square of the singular value of $R_j^l$ with respect to $u_i^l$. (2) (**new bases**) We perform SVD on $\hat{R}_j^l = R_j^l - R_j^l M_j^l (M_j^l)'$ to generate new bases beyond $M_j^l$, i.e., $\hat{R}_j^l = \hat{U}_j^l \hat{\Sigma}_j^l (\hat{V}_j^l)'$ with singular values $\{\hat{\sigma}_{j,h}^l\}_h$. (3) (**select the most important bases from both old and new bases**) Next we concatenate $\{\delta_i^l\}_i$ and $\{(\hat{\sigma}_{j,h}^l)^2\}_h$ together in a vector $\boldsymbol{\delta}$, and sort them in a descending order. We perform $k_j^l$-rank matrix approximation of $R_j^l$, such that the summation of the first $k_j^l$ elements in $\boldsymbol{\delta}$ is greater than $\epsilon_{th}^l \|R_j^l\|_F^2$. Then $B_j^l$ can be constructed by selecting the bases corresponding to the first $k_j^l$ elements in $\boldsymbol{\delta}$.

## 4.4 CONTINUAL LEARNING WITH TRUST REGION GRADIENT PROJECTION

Building on the three modules proposed earlier, i.e., task subspace construction, trust region and scaled weight projection, we next present our approach TRGP for continual learning that efficiently facilitate forward knowledge transfer without forgetting the old tasks.

**Learning task 1.** The first task is learnt using standard gradient descent. The subspace $\{S_1^l\}_{l=1}^L$ is constructed by following Section 4.3.

**Learning task 2, ..., T.** For task $t \in [2, T]$, we first determine the trust region $\mathcal{TR}_t^l$ with top-$K$ correlated old tasks selected for layer $l$. The optimization problem for task $t$ is as follows:

$$\min_{\{W^l\}_l, \{Q_{j,t}^l\}_{l,j \in \mathcal{TR}_t^l}} \mathcal{L}(\{W_{eff}^l\}_l, \mathbb{D}_t), \tag{8}$$
$$s.t \quad W_{eff}^l = W^l + \sum_{j \in \mathcal{TR}_t^l} [\text{Proj}_{S_j^l}^Q(W^l) - \text{Proj}_{S_j^l}(W^l)], \tag{9}$$

where the gradient for updating $W^l$ is $\nabla_{W^l}\mathcal{L} = \nabla_{W^l}\mathcal{L} - (\nabla_{W^l}\mathcal{L})M_t^l(M_t^l)'$ and $M_t^l$ is the bases of all old tasks as in Section 4.3. The subspace $\{S_t^l\}_{l=1}^L$ is next obtained by following Section 4.3.

Table 1: The averaged accuracy (ACC) and backward transfer (BWT) over all the tasks on different datasets. Note that, Multitask jointly learns all tasks only once in a single network by using the whole dataset, which does not adhere to CL setup.

| Method | PMNIST | | CIFAR-100 Split | | 5-Dataset | | MiniImageNet | |
|---|---|---|---|---|---|---|---|---|
| | ACC(%) | BWT(%) | ACC(%) | BWT(%) | ACC(%) | BWT(%) | ACC(%) | BWT(%) |
| Multitask | 96.70 | - | 79.58 | - | 91.54 | - | 69.46 | - |
| OWM | 90.71 | -1 | 50.94 | -30 | - | - | - | - |
| EWC | 89.97 | -4 | 68.80 | -2 | 88.64 | -4 | 52.01 | -12 |
| HAT | - | - | 72.06 | 0 | 91.32 | -1 | 59.78 | -3 |
| A-GEM | 83.56 | -14 | 63.98 | -15 | 84.04 | -12 | 57.24 | -12 |
| ER_Res | 87.24 | -11 | 71.73 | -6 | 88.31 | -4 | 58.94 | -7 |
| GPM | 93.91 | -3 | 72.48 | -0.9 | 91.22 | -1 | 60.41 | -0.7 |
| Ours (TRGP) | **96.34** | **-0.8** | **74.46** | **-0.9** | **93.56** | **-0.04** | **61.78** | **-0.5** |

## 5 EXPERIMENTAL RESULTS

### 5.1 EXPERIMENTAL SETUP

**Datasets and training details.** We evaluate our method on multiple datasets against state-of-the-art CL methods. **1) PMNIST**. Following (Lopez-Paz & Ranzato, 2017; Saha et al., 2021), we create 10 sequential tasks using different permutations where each task has 10 classes. We use a 3-layer fully-connected network. **2) CIFAR-100 Split**. We split the classes of CIFAR-100 (Krizhevsky et al., 2009) into 10 group, and consider 10-way multi-class classification in each group as a single task. Similar with (Serra et al., 2018; Saha et al., 2021), we use a version of 5-layer AlexNet. **3) CIFAR-100 Sup**. We divide the CIFAR-100 dataset into 20 tasks where each task has 5 classes. We use a modified version of LeNet-5. **4) 5-Datasets**. We use a sequence of 5-Datasets which includes CIFAR-10, MNIST, SVHN (Netzer et al., 2011), not-MNIST (Bulatov, 2011) and Fashion MNIST (Xiao et al., 2017), where each dataset is set to be a task. We adapt a reduced ResNet18 network that is used in (Lopez-Paz & Ranzato, 2017). **5) MiniImageNet Split**. We split the 100 classes of MiniImageNet (Vinyals et al., 2016) into 20 sequential tasks where each task has 5 classes, and consider a reduced ResNet18 network. In addition, for all the experiments, the threshold $\epsilon^l$ is set to 0.5, and we select top-2 tasks that satisfy condition Eq. (3). We use the same threshold $\epsilon^l_{th}$ as GPM (Saha et al., 2021) for subspace construction. More details are in the appendix.

**Methods for comparison.** To test the efficacy of our method, we compare it with state-of-the-art approaches in three categories: **1) Memory-based methods**. We compare with Experience Replay with reservoir sampling (ER_Res) (Chaudhry et al., 2019), Averaged GEM (A-GEM) (Chaudhry et al., 2018b), Orthogonal Weight Modulation (OWM) (Zeng et al., 2019) and Gradient Projection Memory (GPM) (Saha et al., 2021). **2) Regularization-based methods**. We compare with state-of-the-art HAT (Serra et al., 2018) and Elastic Weight Consolidation (EWC) (Kirkpatrick et al., 2017). **3) Expansion-based methods**. We further compare with Progressive Neural Network (PNN) (Rusu et al., 2016), Learning Without Forgetting (LWF) (Li & Hoiem, 2017), Dynamic-Expansion Net (DEN) (Yoon et al., 2017), and APD (Yoon et al., 2020), by using CIFAR-100 Sup dataset.

**Metrics.** Following GPM (Saha et al., 2021), two metrics are used to evaluate the performance: Accuracy (ACC), the average final accuracy over all tasks, and Backward Transfer (BWT), which measures the forgetting of old tasks when learning new tasks. ACC and BWT are defined as:

$$ACC = \frac{1}{T} \sum_{i=1}^{T} A_{T,i}, BWT = \frac{1}{T-1} \sum_{i=1}^{T-1} A_{T,i} - A_{i,i} \qquad (10)$$

where $T$ is the number of tasks, $A_{T,i}$ is the accuracy of the model on $i$-th task after learning the $T$-th task sequentially.

### 5.2 MAIN RESULTS

**ACC and BWT comparison.** As shown in Table 1, TRGP achieves significantly accuracy improvement compared with prior works on all datasets. For example, in contrast to the best prior results, TRGP achieve the accuracy gain of 2.43%, 1.98% and 1.37% over GPM on PMNIST, CIFAR-100 Split and MiniImageNet, respectively, and 2.34% over HAT on 5-Dataset. Surprisingly, we could even achieve better accuracy than Multitask on 5-Datasets, which usually serves as an upper bound for CL benchmarks. This superior performance of TRGP clearly shows its capability to effectively facilitate forward knowledge transfer. In addition, TRGP also demonstrates strong performance with the lowest BWT, reducing 0.2% than OWM and 0.6% than GPM, even with 5.63% and 2.34% ac-

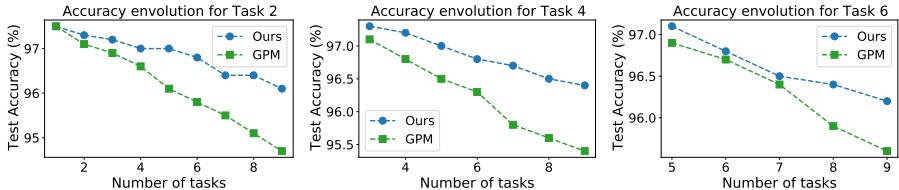

Figure 4: The final accuracy for all tasks on three datasets (GPM VS Ours).

Table 2: The performance for CIFAR-100 Sup dataset. Note that Single-task learning (STL) trains a separate network for each task, which does not adhere to CL setup.

| Metric | Methods | | | | | | |
|---|---|---|---|---|---|---|---|
| | STL | PNN | DEN | RCL | APD | GPM | Ours (TRGP) |
| ACC(%) | 61.00 | 50.76 | 51.10 | 51.99 | 56.81 | 57.72 | **58.25** |
| Capacity(%) | 2000 | 271 | 191 | 184 | 130 | 100 | 100 |

curacy improvement on PMNIST and 5-Dataset, respectively. Compared with HAT on CIFAR-100 Split, TRGP has marginally worse BWT, but achieves 2.4% accuracy gain.

Moreover, TRGP exhibits an universal dominance over GPM about the final accuracy of all tasks on all the three datasets. According to the apple to apple comparison with GPM in Fig. 4, one interesting phenomenon is observed: TRGP has the similar accuracy on "easy" tasks, but significantly improves the accuracy on the "difficult" tasks. For example, in the 5-Dataset setting, both TRGP and GPM achieve good accuracy on Task 1 (MNIST) and 3 (Fashion MNIST), which can be easily trained well, but TRGP significantly outperforms GPM on the rest three more difficult Tasks (CIFAR-10, SVHN and NotMNIST). In the end, as shown in Table 2, we further compare with the expansion-based methods by using CIFAR-100 Sup setting. It can be seen that TRGP outperforms all other CL methods, with a fixed capacity network.

**Discussion.** We next show the accuracy evolution of specific tasks during the training of all tasks sequentially. We randomly select three tasks for each dataset to compare with GPM (we only show results on PMNIST in Fig. 5 and relegate the rest to the appendix). There are two main obervations: 1) TRGP completely outperforms GPM during training for all the sequential tasks on the three datasets; 2) For the PMNIST and 5-Dataset settings, TRGP could significantly reduce forgetting. To understand why, consider the case where GPM and TRGP learns a new task $t$ given the same model $\mathbb{W}_{t-1}$, and denote $\{M_{t-1}^l\}_l$ as the bases of all old tasks. Then we can have

*For GPM,* the effective weight for layer $l$ is
$$\boldsymbol{W}_{eff}^l = \text{Proj}_{\boldsymbol{M}_{t-1}^l}(\boldsymbol{W}^l) + \text{Proj}_{\perp \boldsymbol{M}_{t-1}^l}(\boldsymbol{W}^l) \tag{11}$$
where the weight projection on $\boldsymbol{M}_{t-1}^l$ is frozen to protect old tasks, and only the weight projection orthogonal to $\boldsymbol{M}_{t-1}^l$ can be updated for learning task $t$.

*For TRGP,* the effective weight for layer $l$ is
$$\boldsymbol{W}_{eff}^l = \text{Proj}_{\{S_j^l\}_{j \notin \mathcal{TR}_t^l}}(\boldsymbol{W}^l) + \text{Proj}_{\{S_j^l\}_{j \in \mathcal{TR}_t^l}}^Q(\boldsymbol{W}^l) + \text{Proj}_{\perp \boldsymbol{M}_{t-1}^l}(\boldsymbol{W}^l) \tag{12}$$
where only the first term, i.e., weight projection on subspaces of old tasks that are not in the trust region $\mathcal{TR}_t^l$, is frozen for task $t$. In contrast to GPM, an additional and also important part of weights, i.e., the scaled weight projection on subspaces of related old tasks in $\mathcal{TR}_t^l$, can be learnt in a favorable way for task $t$. As a result, TRGP can achieve better forward knowledge transfer by explicitly and cleverly reusing the important knowledge of strongly correlated old tasks in the trust region. More interestingly, benefiting from the task-unique information captured by the scaled weight projection, the backward transfer can also be reduced.

Figure 5: Accuracy evolution for different tasks on PMNIST setting.

Table 3: Ablation study on CIFAR-100 Split and 5-Datasets settings.

| Datasets | Impact of threshold $\epsilon^l$ | | | Layer-wise VS Task-wise | | Number of selected tasks | |
|---|---|---|---|---|---|---|---|
| | 0.2 | 0.5 | 0.7 | Layer-wise | Task-wise | Top-1 | Top-2 |
| CIFAR-100 | 74.52 | 74.46 | 74.30 | 74.46 | 73.25 | 74.00 | 74.46 |
| 5-Datasets | 93.28 | 93.56 | 93.43 | 93.56 | 92.85 | 92.94 | 93.56 |

## 5.3 ABLATION STUDY AND ANALYSIS

**Impact of the threshold $\epsilon^l$.** To understand the impact of the threshold $\epsilon^l$, we evaluate the learning performance for three different values of $\epsilon^l$ (i.e., 0.2, 0.5, 0.7) as shown in Table 3. The results show that the accuracy is very stable across the three threshold values, with ignoble accuracy difference on both CIFAR-100 Split and 5-Dataset settings. The reason behind is because we only select top-2 old tasks with largest gradient projection norm into the trust region, among all tasks satisfying condition Eq. (3). Therefore, for a wide range of $\epsilon^l$, the selected tasks in the trust region are actually fixed. The small accuracy fluctuation is because with some possibility only one old task satisfies Eq. (3) and is selected for some layers when $\epsilon^l$ increases. Overall, TRGP is very robust to the value of $\epsilon^l$.

**Layer-wise *vs*. Task-wise trust region.** To show the efficacy of layer-wise trust region, we compare it with the task-wise variant which shares a fixed trust region across all layers for each task. First, as shown in Table 3, layer-wise could achieve 1.21% accuracy gain over task-wise on CIFAR-100 Split. Furthermore, we illustrate the final accuracy of all tasks for layer-wise and task-wise of the proposed TRGP, and GPM on CIFAR-100 Split setting in Fig. 6. First, the performance of layer-wise is better than or comparable to task-wise for all tasks, because layer-wise provides a much finer characterization of task correlations in terms of layer-level features. Then, it is interesting to see that the learning behavior for the three cases follows the same trend. This observation further corroborates that TRGP can improve the accuracy and mitigate forgetting on both "easy" and "difficult" tasks.

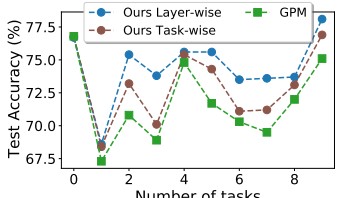

Figure 6: The final accuracy for all tasks of Task-wise *VS* Layer-wise on CIFAR-100 Split.

**Impact of selected tasks in trust region.** We first evaluate the accuracy of Top-1 and Top-2 selected tasks as shown in Table 3. It shows that selecting the top-2 most correlated tasks could achieve better accuracy on both CIFAR-100 Split and 5-Dataset settings. Note that good performance can also be achieved even with the Top-1 case. Moreover, we illustrate the detailed task selection in the trust region for both layer-wise and task-wise on 5-Dataset setting in Fig. 7. For the task-wise, current task always selects the two adjacent previous tasks for all layers. Differently, the task selection varies for layer-wise, leading to more accurate selection of related tasks for each layer. For example, the layer wise trust region for Task 4 (Fashion MNIST) selects Task 1 (MNIST) or Task 3 (not-MNIST) as the most related tasks almost for all layers, over Task 0 (CIFAR-10) and Task 2 (SVHN), which clearly makes sense because Fashion MNIST shares more common features with MNIST and not-MNIST.

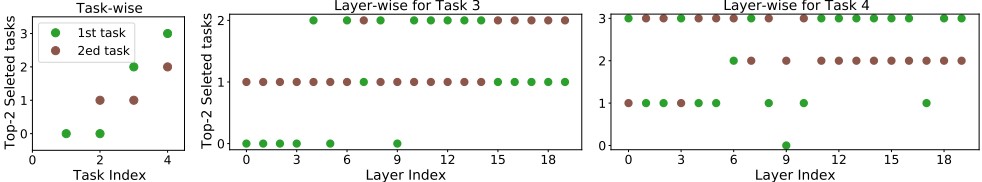

Figure 7: The detailed selected tasks on 5-Datasets setting.

## 6 CONCLUSION

In this work, we propose trust region gradient projection for continual learning to facilitate forward knowledge transfer with forgetting, based on an efficient characterization of task correlation. Particularly, our approach is built on two key blocks, i.e., the layer-wise trust region which effectively select the old tasks strongly correlated to the new task in a single-shot manner, and scaled weight projection which cleverly reuses the frozen weights of old tasks in the trust region without modifying the model. Extensive experiments show that our approach significantly improves over the related state-of-the-art methods.

ACKNOWLEDGEMENT

This work is supported in part by NSF Grants CNS-2003081, CNS-2203239, CPS-1739344, and CCSS-2121222.

REPRODUCIBILITY STATEMENT

For the experimental results presented in the main text, we include the code in the supplemental material, and specify all the training details in Section 5.1 and Appendix A. For the datasets used in the main text, we also give a clear explanation in Section 5.1.

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

# A EXPERIMENT SETUPS

**Training hyper-parameters.** We evaluate our method on multiple datasets against state-of-the-art continual learning methods. **1) PMNIST**. We use a 3-layer fully-connected network. with two hidden layer of 100 units. and train the network for 5 epochs with batch size of 10 for each task. **2) CIFAR-100 Split**. CIFAR-100 (Krizhevsky et al., 2009) consists of images from 100 generic object classes. We use a version of 5-layer AlexNet and train each task for maximum of 200 epochs with the early termination strategy based on the validation loss value. The batch size is set to 64. **3) CIFAR-100 Sup**. We use a modified version of LeNet-5 with 20-50-800-500 neurons and train 50 epochs for each task sequentially. The batch size is set to 64. **4) 5-Datasets**. We train each task for maximum of 200 epochs with the early termination strategy. The batch size is set to 64. **5) MiniImageNet Split**. Following GPM (Saha et al., 2021), we use the reduced ResNet18 architecture, where the covolution with stride 2 in the first layer. We train each task for maximum of 100 epochs with the early termination strategy with 0.1 initial learning rate and 64 batchsize. In addition, for all the experiments, the threshold $\epsilon^l$ is set to 0.5, and we select top-2 tasks that satisfy condition Eq. (3). We use the same threshold $\epsilon_{th}^l$ as GPM (Saha et al., 2021) for subspace construction. We initialize the scaling matrix with the identity matrix and train all models with plain stochastic gradient descent.

# B MORE EXPERIMENTAL RESULTS

## B.1 ACCURACY EVOLUTION

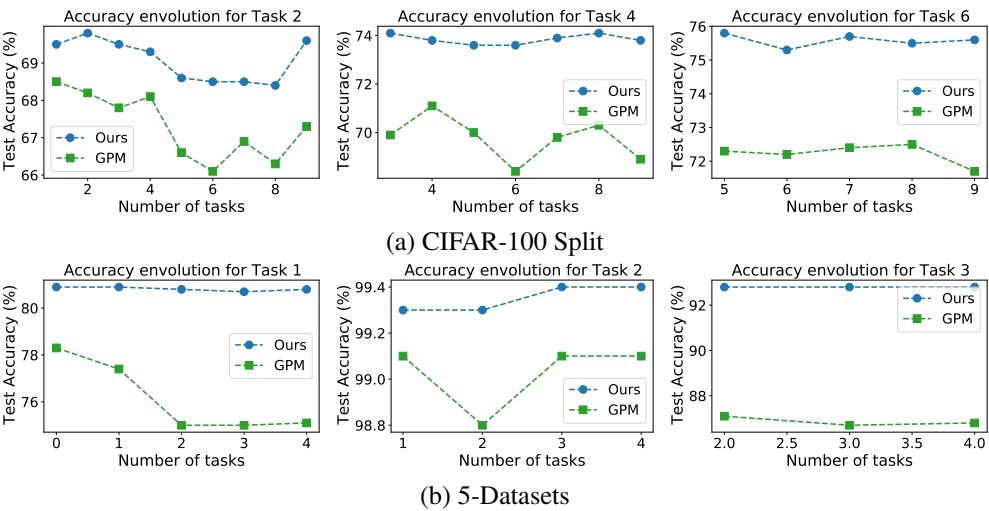

(a) CIFAR-100 Split

(b) 5-Datasets

Figure 8: Accuracy evolution for different tasks on CIFAR-100 Split and 5-Datasets settings.

## B.2 STANDARD DEVIATION

We have summarized the results on the standard deviation for the averaged accuracy and backward transfer over 5 different runs on all datasets in Table 4.

Table 4: The averaged accuracy (ACC) and backward transfer (BWT) with the standard deviation values over 5 different runs on different datasets.

| Method | PMNIST | | CIFAR-100 Split | | 5-Dataset | | MiniImageNet | |
|---|---|---|---|---|---|---|---|---|
| | ACC(%) | BWT(%) | ACC(%) | BWT(%) | ACC(%) | BWT(%) | ACC(%) | BWT(%) |
| Multitask | $96.70 \pm 0.02$ | - | $79.58 \pm 0.54$ | - | $91.54 \pm 0.28$ | - | $69.46 \pm 0.62$ | - |
| OWM | $90.71 \pm 0.11$ | $-1 \pm 0$ | $50.94 \pm 0.60$ | $-30 \pm 1$ | - | - | - | - |
| EWC | $89.97 \pm 0.57$ | $-4 \pm 1$ | $68.80 \pm 0.88$ | $-2 \pm 1$ | $88.64 \pm 0.26$ | $-4 \pm 1$ | $52.01 \pm 2.53$ | $-12 \pm 3$ |
| HAT | - | - | $72.06 \pm 0.50$ | $0 \pm 0$ | $91.32 \pm 0.18$ | $-1 \pm 0$ | $59.78 \pm 0.57$ | $-3 \pm 0$ |
| A-GEM | $83.56 \pm 0.16$ | $-14 \pm 1$ | $63.98 \pm 1.22$ | $-15 \pm 2$ | $84.04 \pm 0.33$ | $-12 \pm 1$ | $57.24 \pm 0.72$ | $-12 \pm 1$ |
| ER_Res | $87.24 \pm 0.53$ | $-11 \pm 1$ | $71.73 \pm 0.63$ | $-6 \pm 1$ | $88.31 \pm 0.22$ | $-4 \pm 0$ | $58.94 \pm 0.85$ | $-7 \pm 1$ |
| GPM | $93.91 \pm 0.16$ | $-3 \pm 0$ | $72.48 \pm 0.40$ | $-0.9 \pm 0$ | $91.22 \pm 0.20$ | $-1 \pm 0$ | $60.41 \pm 0.61$ | $-0.7 \pm 0.4$ |
| Ours (TRGP) | $\mathbf{96.34 \pm 0.11}$ | $\mathbf{-0.8 \pm 0.1}$ | $\mathbf{74.46 \pm 0.32}$ | $\mathbf{-0.9 \pm 0.01}$ | $\mathbf{93.56 \pm 0.10}$ | $\mathbf{-0.04 \pm 0.01}$ | $\mathbf{61.78 \pm 0.60}$ | $\mathbf{-0.5 \pm 0.6}$ |

## B.3 Forward transfer

To evaluate the forward transfer, we follow the metric used in (Veniat et al., 2020) and consider the accuracy of the model on $i$-th task after learning the $i$-th task sequentially, i.e., $A_{i,i}$ as defined in Eq. (10). Tables 5 - 8 summarize the comparison of $A_{i,i}$ for each task $i$ between GPM and TRGP on PMNIST, CIFAR-100 Split and 5-Dataset, respectively. As the same baseline (e.g., the accuracy of the model learnt from scratch using the task's own data) for each task will be used when evaluating the forward transfer for GPM and TRGP, we can infer that TRGP achieves the forward transfer gain of 0.17%, 2.01%, 2.00% and 2.36% over GPM on PMNIST, CIFAR-100 Split, 5-Datasets and MiniImageNet respectively.

Table 5: The accuracy $A_{i,i}$ of the model on $i$-th task after learning the $i$-th task sequentially on PMNIST 10 tasks.

| Methods | 1 | 2 | 3 | 4 | 5 | 6 | 7 | 8 | 9 | 10 | Avg |
|---|---|---|---|---|---|---|---|---|---|---|---|
| GPM | 97.5 | 97.5 | 97.3 | 97.1 | 97.0 | 96.9 | 96.8 | 96.4 | 96.5 | 96.5 | 96.95 |
| Ours (TRGP) | **97.5** | **97.5** | **97.5** | **97.3** | **97.1** | **97.1** | **96.9** | **96.7** | **96.9** | **96.7** | **97.12** |

Table 6: The accuracy $A_{i,i}$ of the model on $i$-th task after learning the $i$-th task sequentially on CIFAR-100 Split 10 tasks.

| Methods | 1 | 2 | 3 | 4 | 5 | 6 | 7 | 8 | 9 | 10 | Avg |
|---|---|---|---|---|---|---|---|---|---|---|---|
| GPM | 76.8 | 68.5 | 72.4 | 69.9 | 74.8 | 72.3 | 70.3 | 71.9 | 73.2 | 75.1 | 72.52 |
| Ours (TRGP) | **76.9** | **69.5** | **75.1** | **74.1** | **75.3** | **75.8** | **72.8** | **73.8** | **73.9** | **78.1** | **74.53** |

Table 7: The accuracy $A_{i,i}$ of the model on $i$-th task after learning the $i$-th task sequentially on 5-Dataset 5 tasks.

| Methods | 1 | 2 | 3 | 4 | 5 | Avg |
|---|---|---|---|---|---|---|
| GPM | 78.3 | 99.1 | 87.1 | 99.1 | 94.1 | 91.54 |
| Ours (TRGP) | **80.9** | **99.3** | **92.8** | **99.4** | **95.3** | **93.54** |

Table 8: The accuracy $A_{i,i}$ of the model on $i$-th task after learning the $i$-th task sequentially on MiniImageNet Split 20 tasks.

| Methods | 1 | 2 | 3 | 4 | 5 | 6 | 7 | 8 | 9 | 10 | 11 | 12 | 13 | 14 | 15 | 16 | 17 | 18 | 19 | 20 | Avg |
|---|---|---|---|---|---|---|---|---|---|---|---|---|---|---|---|---|---|---|---|---|---|
| GPM | 58.6 | 63.6 | 57.2 | 59.0 | 53.6 | 78.0 | 63.0 | 66.0 | 74.0 | 83.8 | 43.0 | 60.4 | 55.6 | 57.8 | 59.6 | 53.0 | 56.0 | 47.6 | 66.0 | 56.8 | 60.63 |
| Ours (TRGP) | **58.7** | **66.1** | **59.2** | **59.3** | **57.1** | **81.4** | **67.3** | **70.1** | **75.7** | **85.2** | **43.2** | **61.8** | **58.0** | **60.1** | **60.0** | **54.8** | **61.4** | **48.4** | **69.8** | **62.2** | **62.99** |

## B.4 Computational complexity

Memory: In terms of the memory, the major difference between TRGP and GPM is that TRGP requires additional memory to store the scaling matrices for each task. However, since the dimension of the scaling matrix is the same with the number of the extracted bases for the input subspace, which is usually small and controllable by the matrix approximation accuracy $\epsilon_{th}$ in Eq. (7), the memory increase is marginal and controllable. Particularly, the memory usage of TRGP can be further reduced by only learning the scaling matrices for the convolutional layers.

Training time: We compare the training time between TRGP and other baselines on relatively complex task sequences. As shown in Table 9, for CIFAR-100 Split, TRGP takes around 65% more time than GPM, is comparable with HAT and ER_Res, and takes less time than OWM and EWC; for 5-Datasets, TRGP takes around 21% more time than GPM, but is much faster than other baselines including EWC, HAT, A-GEM and ER_Res; for MiniImageNet, TRGP tasks around 34% more time than GPM, is comparable with EWC, but is much faster than A-GEM.

Table 9: Training time comparison on CIFAR-100 Split, 5-Datasets and MiniImageNet. Here the training time is normalized with respect to the value of GPM. Please refer (Saha et al., 2021) for more specific time.

| Dataset | Methods | | | | | | |
|---|---|---|---|---|---|---|---|
| | OWM | EWC | HAT | A-GEM | ER_Res | GPM | Ours (TRGP) |
| CIFAR-100 | 2.41 | 1.76 | 1.62 | 3.48 | 1.49 | 1 | 1.65 |
| 5-Datasets | - | 1.52 | 1.47 | 2.41 | 1.40 | 1 | 1.21 |
| MiniImageNet | - | 1.22 | 0.91 | 1.79 | 0.82 | 1 | 1.34 |

### B.5  ACCURACY VS LEARNING EPOCHS

The learning dynamics for each task are shown in Figure 9 and 10. Clearly, our approach can perform significantly better than GPM on some tasks, especially for the tasks in the tail of the task sequence. This is because in GPM, with more tasks being learnt, the optimization space for new tasks becomes more restrictive, leading to limited performance for new tasks. Note that the y-axis is the validation accuracy with a split validate dataset that used during training, by following the setup in (Saha et al., 2021). The validation accuracy varies because the size of the validate dataset is relatively small (See (Saha et al., 2021) for the specific size). For the testing accuracy in all the tables, we evaluate the accuracy with the testing dataset after training.

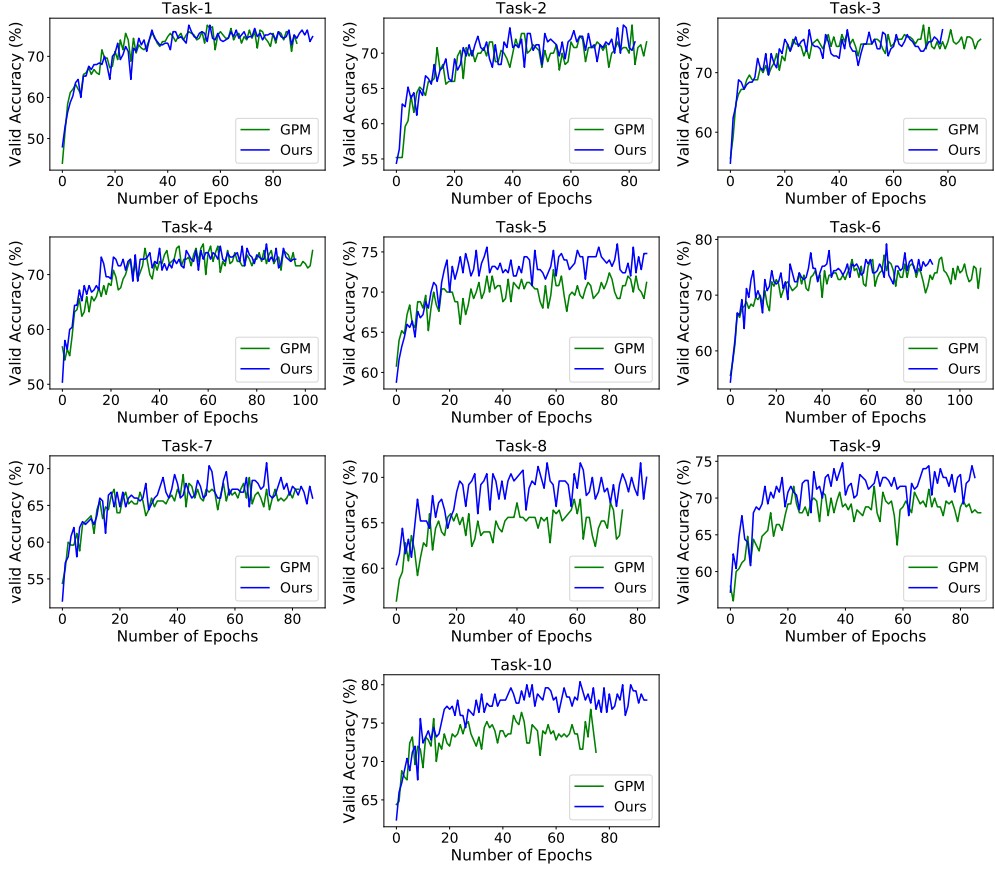

Figure 9: Accuracy vs learning epochs for different tasks on CIFAR-100 Split.

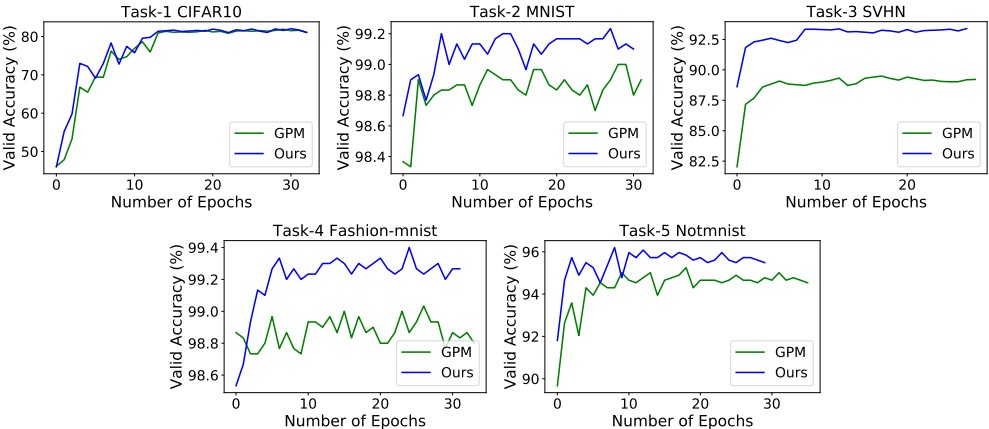

Figure 10: Accuracy vs learning epochs for five tasks on 5-Dataset.

