# OpenReview forum: "TRGP: Trust Region Gradient Projection for Continual Learning"
_ICLR.cc/2022/Conference — ICLR 2022 Spotlight_

### Official Review · Reviewer_tgxV · 2021-11-02

**Correctness:** 2
**Technical Novelty And Significance:** 2
**Empirical Novelty And Significance:** 2
**Recommendation:** 3
**Confidence:** 4

**Main Review:**

## Strengths
- This paper points out a major issue of GPM (Saha et al.): the orthogonal projection is too restrictive.
- The proposed heuristic empirically improves the performance.

## Weaknesses
### The writing is hard to follow
Since this paper introduces multiple new concepts, it was hard for me to understand the overall algorithm and the intuition behind it.
I think the writing can be improved a lot.
For example,
- It was hard to find out what the authors are trying to do with the trust region. I was confused if they are going to allow or restrict the model update in the trust region.
- In Figure 2, $S_l^f$ seems to be the 2D input subspace of layer $l$ in 3D input space. But why is there $\nabla_{W^l} \mathcal L_t$, which should be in the parameter space? According to Eq.(2), $\nabla_{W^l} \mathcal L_t$ and $\mathrm{Proj} (\nabla_{W^l} \mathcal L_t)$ are matrices, but why are they represented as a vector?

### Some of the reasoning steps are controversial
- Toy example 1 is not a good example. The authors claim that orthogonal projection is problematic because the optimal model for task 2 should be $W_2^l = -W_1^l$. However, that will lead to complete forgetting of the first task. In fact, this example is impossible to solve with a single linear layer.
- Therefore, I do not agree with the conjecture motivated by this example: *naive orthogonal projection could possibly compromise the learning performance of the new task that is strongly correlated with old tasks, especially when the correlation is “negative” as in the toy example 1*. Toy example 1 is a situation where the model has to choose between two conflicting options: (i) perfectly memorize task 1 and ignore task 2, (ii) completely forget task 1 and learn task 2. Orthogonal projection is a method for option (i), which is one of the best things a linear model can do.
- Considering my previous arguments, the claim that the weight projection to the subspace of trust region should be modified is somewhat arbitrary. I think more theoretical justification is needed.

### Lack of justification and empirical evidence for the scaled weight projection
Although the authors claim scaled weight projection as one of their main contributions, there is no explanation of how the scaling matrix $Q_{j,t}^l$ in Eq.(6) is initialized and trained.
Also, there is no ablation study to prove its effectiveness.
Without further information, I cannot acknowledge this component as a contribution.

### Limited novelty
In the end, the authors perform a relaxed orthogonal projection of the gradient.
The proposed method is just one way of relaxing the orthogonal projection, and there is not enough justification for why this particular algorithm should be effective.
I think there can be various ways to relax the orthogonal projection.
It would be better if the authors could provide a thorough empirical analysis of multiple relaxing schemes or theoretical justification for the proposed method.

**Summary Of The Paper:**

This paper proposes a continual learning method based on gradient projection memory (GPM) of Saha et al., which projects the gradient of each layer to be orthogonal to the input subspace of previous tasks.
Motivated by the fact that the orthogonal projection can harm the performance by being too restrictive, the authors propose a heuristic algorithm to reduce the restriction.
Specifically, the authors choose a subset of most "correlated" tasks and let the model change along the subspace of the correlated tasks.

**Summary Of The Review:**

This paper proposes a heuristic to improve GPM (Saha et al.).
However, the proposed method lacks theoretical justification, useful insights, and some essential experiments.
Also, the overall writing should be improved.
Therefore, I do not think this paper meets the ICLR standards.

---

> ### Author Response · Authors · 2021-11-22
> **Reply to Reviewer tgxV (2/2)**
>
> Q4: Explanation of the scaling matrix.
>
> A4: We have the following clarifications about the scaling matrix.
>
> - We have clearly stated the optimization about the scaling matrix $Q$ in the optimization problem Eq. (8) in section 4.4 when we summarize TRGP. It can be seen that the scaling matrix $Q$ is jointly optimized with the network weight $W$. In the experiments, we initialize $Q$ with the identity matrix.
>
> - As we presented throughout the entire study, our approach learns the new task by i) efficiently selecting the most correlated old tasks to the trust region of the new task and ii) reusing the weight projection of the selected old tasks by learning a scaling matrix to prompt the forward knowledge transfer. One key technical difference between TRGP and GPM is that TRGP introduces a scaling matrix to scale weight projections of the selected old tasks in the trust region, as clearly shown in Eq. (11) and Eq. (12). Therefore, the entire section of the experimental study is to demonstrate the effectiveness of learning a scaling matrix to scale the weight projections of the selected old tasks in the trust region.
>
>
> Q5: In the end, the authors perform a relaxed orthogonal projection of the gradient.
>
> A5: We kindly disagree with your claim that our approach performs a relaxed orthogonal projection of the gradient. We have never used the terminology "relaxed orthogonal projection" or claimed that TRGP performs a relaxed orthogonal projection of the gradient throughout the entire paper. In fact, our approach does not relax orthogonal projection but compensates it by facilitating the forward knowledge transfer.

---

> > ### Author Response · Authors · 2021-11-27
> > **Reply to Reviewer tgxV**
> >
> > We thank the reviewer again for the thorough reviews and comments. Since the final stage of the discussion will end soon, please let us know if you have further questions on our response, and we will be more than happy to answer your questions.

---

> ### Author Response · Authors · 2021-11-22
> **Reply to Reviewer tgxV (1/2)**
>
> Thanks you for your thorough reviews and constructive comments. Per your suggestions, we have made the following major revision:  (1) added new experiments on Split-MiniImageNet in Table 1 in section 5.2, (2) added the standard deviation in Table 4 in Appendix B.2, (3) added the forward transfer comparison in Table 5-8 in Appendix B.3, (4) added training time comparison in Table 9 in Appendix B.4, (5) added the learning curves in Figure 9 and 10 in Appendix B.5, and (6) made various revisions throughout the paper based on all reviewers’ comments. All our changes are highlighted with blue-colored texts. New comments on these changes are very welcome!
>
>
> Q1: It was hard to find out what the authors are trying to with trust region.
>
> A1: We have clearly stated the main ideas behind our approach at the beginning of Section 4: i) we first introduce a novel notion of trust region to select the most related old tasks for the new task efficiently in a one-shot manner; ii) we next reuse the weight projection of the selected old tasks in the trust region through a scaling matrix to facilitate forward knowledge transfer without forgetting.
>
> - Particularly, in Section 4.1, right after Eq. (2) we have emphasized again that the trust region for a new task is defined as a set of its most related old tasks and selected based on Eq. (3).
>
> - And in the third paragraph in section 4.2,  we have clearly explained the intuition of how to reuse the weight projection of the old tasks in the trust region:
> (1) we first note that the subspaces of the related old tasks in the trust region are very likely to contain important bases for the new task, and the weight projection in a subspace is indeed a linear combination of the weight projection onto bases in the subspace; (2) therefore, the weight for the new task can be obtained by scaling the weight projection of the most correlated old tasks onto each basis through a scaling matrix $Q$ as shown in Eq. (6), and then we optimize $Q$ to find the best scaling coefficients for the new task as summarized in the optimization problem Eq. (8) in section 4.4. As clearly shown in this optimization problem, the weight projections for the old tasks are not modified when learning the new task because of the orthogonal gradient projection, and $Q$ is optimized to appropriately scale the weight projections of selected old tasks in the trust region.
>
>
>
> Q2: According to Eq 2, $\nabla_{W^l} L_t$ and $proj(\nabla_{W^l} L_t)$ are matrices, but why are they represented as a vector in Figure 2?
>
> A2: As we clearly stated in the first paragraph after Definition 1, Figure 2 is just used to illustrate a simple example of the trust region where the subspace is 2-dimensional and the weight here is a vector. In contrast, Eq. (2) is the formal definition of the matrix projection onto any subspace.
>
>
>
> Q3: Conjecture motivated by the toy example 1.
>
> A3: We have the following clarifications for the toy example 1.
>
> - You are right about that orthogonal projection perfectly learns task 1 but ignores task 2 in the toy example 1. However, this is exactly the point we try to make in example 1 about the potential drawback of the orthogonal projection and continual learning needs to go beyond orthogonal projection. It is very clear that the orthogonal projection comprises the learning performance of task 2 in order to address the forgetting of task 1. You can think this as one of the best things a linear model can do with orthogonal projection, but this is not the best thing that continual learning should do.
>
> - Since these two tasks are highly correlated, intuitively there should be better forward knowledge transfer between them. In fact, a simple yet effective solution for the toy example 1 is to reuse the weight and multiply it by a scaling matrix with all -1 elements. This motivates our approach about effectively selecting the most correlated old tasks and then reusing their weight projections through a scaling matrix to facilitate the forward knowledge transfer.
>
> - Our approach does not modify the weight projection of selected old tasks in the trust region. In fact, our approach freezes the weight projection of the old tasks to address catastrophic forgetting, but learns a scaling matrix to reuse and scale the weight projection of the most correlated old tasks to facilitate forward knowledge transfer.

---

### Official Review · Reviewer_aL1A · 2021-11-02

**Correctness:** 3
**Technical Novelty And Significance:** 3
**Empirical Novelty And Significance:** 2
**Recommendation:** 6
**Confidence:** 4

**Main Review:**

Strengths:

    1. The problem of forward knowledge transfer is an unexplored problem in the continual learning literature.

    2. The paper reads well.

    3. Experiments are somewhat convincing.

Weaknesses:

    1. Experiments are not extensive enough and important benchmarks are missing.

    2. Thorough comparison with recent works is missing.



**Summary Of The Paper:**

In this paper, the problem of forward knowledge transfer in continual learning settings is explored. The idea is to measure correlations between the learned tasks based on the notion of "trust region" which helps to identify the most similar learned tasks to the current task. The core idea is that frozen weights for similar past tasks can be relaxed to reuse them to learn the current task. Since task similarities are used for this purpose, this will not lead to catastrophic forgetting and at the same time helps to transfer knowledge. Experiments on four benchmarks are provided to demonstrate that the method is effective.

**Summary Of The Review:**

Current works in continual learning focus mostly on tackling catastrophic forgetting and overlook accumulative learning. Hence, I think the authors have selected a good area and the proposed idea also seems to be sound. However, I have some reservations about this work:

1. Given the expectation in the recent literature on continual learning, only relatively simple datasets are considered in the experiments. I think results on more complex datasets such as split-Sub-ImageNet and Split-ImageNet should be added.

2. I was wondering why forward transfer metric is not used for comparison? I think it is as informative as backward transfer metric.

3.  An area of interest is to analyze the learning curves and dynamics of learning, i.e., performance vs learning epoch. It is helpful to see if the proposed method enables better jumpstart performance when a new task is learned.

4. Comparison with more works is missing. For example, EWC and HAT are both outdated for regularization-based methods and many recent follow-ups exist. For a realistic comparison against prior works, state-of-the-art methods for each group of methods should be included. Please check the recent literature.

5. Computational complexity is overlooked. Whereas it is an important factor for continual learning. I think it is necessary to include how much additional computational load should be performed to benefit from TRGP.

6. For a more informative comparison, the standard deviation in results should be reported. Also, BWT metric can be reported more accurately by including more decimals.

 In conclusion, I think this work is in a good direction but further improvement is necessary to make it suitable for a venue similar to ICLR.

---

> ### Author Response · Authors · 2021-11-22
> **Reply to Reviewer aL1A**
>
> Thank you for your thorough reviews and constructive comments. Per your suggestions, we have made the following major revision:  (1) added new experiments on Split-MiniImageNet in Table 1 in section 5.2, (2) added the standard deviation in Table 4 in Appendix B.2, (3) added the forward transfer comparison in Table 5-8 in Appendix B.3, (4) added training time comparison in Table 9 in Appendix B.4, (5) added the learning curves in Figure 9 and 10 in Appendix B.5, and (6) made various revisions throughout the paper based on all reviewers’ comments. All our changes are highlighted with blue-colored texts. New comments on these changes are very welcome!
>
> Q1: Results on more complex datasets should be added.
>
> A1: Per the reviewer's suggestion, we have added the experiments on Split-MiniImageNet in Table 1 in section 5.2, by following the recent studies, e.g., Saha et al. 2021, Chaudhry et al. 2019. It can be clearly seen from Table 1 that TRGP achieves the best performance and improves over the state-of-the-art approach GPM.
>
> Q2: Forward transfer metric.
>
> A2: In terms of the forward transfer metric, we have the following clarifications.
>
> - To the best of our knowledge, there is currently no standard evaluation metric for forward transfer. For example, D. Lopez-Paz \& MA Ranzato 2017 evaluates the forward transfer for task $t$ as the difference between i) the testing accuracy with data of task $t$ using the model learnt after task $t-1$ and ii) the testing accuracy with data of task $t$ using a randomly initialized model; T. Veniat et al. 2021 calculates the forward transfer as the difference of performance between a model that has learnt through a whole sequence of tasks and a model that hast only learnt the last task. Therefore, following the previous work (e.g., R. Aljundi et al. 2018, A. Chaudhry et al. 2019, Y. Guo et al. 2020, Saha et al. 2021), we have only considered the average accuracy and the backward transfer.
>
> - Per the reviewer's suggestion, in Appendix B.3 we have added the testing accuracy for each task using the model learnt just after this task, and compare the performance between GPM and our approach. This can serve as a surrogate of the forward transfer because all approaches use the same baseline accuracy when evaluating the forward transfer. As shown in Tables 5-8, our approach achieves the forward transfer gain of 0.17\%, 2.01\%, 2.00\%  and 2.36\% over GPM on PMNIST, CIFAR-100 Split, 5-Datasets and Split-MiniImageNet, respectively.
>
>
>
> Q3: Learning curves and dynamics of learning.
>
> A3: Per the reviewer's suggestion, we have added the learning curves in Appendix B.5. As shown in Figure 9 and 10, our approach can perform significantly better than GPM on some tasks, especially for the tasks in the tail of the task sequence. This is because in GPM, with more tasks being learnt, the optimization space for new tasks will become more restrictive, leading to limited performance for the new tasks.
>
>
> Q4: Comparison with more works.
>
> A4: To the best of our knowledge, GPM (Saha et al. 2021), which is published in ICLR 2021, achieves the state-of-the-art performance among non-expansion methods for continual learning. Therefore, we compare our approach with GPM and the baselines therein. We will try to search for more competitive baselines and add the comparison in the final revision.
>
>
> Q5: How much additional computational load should be performed to benefit from TRGP.
>
> A5: Per the reviewer's suggestion, we have added the training time comparison between TRGP and other baselines on relatively complex task sequences in Table 9 in Appendix B.4. For example, for CIFAR-100 Split, TRGP takes around 60\% more time than GPM, is comparable with HAT and ER\_Res, and takes less time than OWM and EWC; for 5-Datasets, TRGP takes around 20\% more time than GPM, but is much faster than other baselines including EWC, HAT, A-GEM and ER\_Res.
>
>
> Q6: Standard deviation.
>
> A6: Per the reviewer's suggestion, we have added the standard deviation over 5 different runs in Table 4 in Appendix B.2.

---

> > ### Comment · Reviewer_aL1A · 2021-11-24
> > **Updated Rate**
> >
> > Thank you for addressing my concerns. I updated my rating accordingly.

---

> > > ### Author Response · Authors · 2021-11-27
> > > **Many thanks for your further updates!**
> > >
> > > We thank the reviewer very much for further reviewing our response and increasing the score!

---

### Official Review · Reviewer_jo3a · 2021-11-02

**Correctness:** 3
**Technical Novelty And Significance:** 4
**Empirical Novelty And Significance:** 3
**Recommendation:** 8
**Confidence:** 3

**Main Review:**


Strengths:
1.This paper put forward the problems of the existing methods and provides a detailed discussion.
2.Propose a novel solution to this problem, which achieves substantial performance improvement on all benchmarks and overwhelmed its baseline method.

Weakness:
1.Toy example 1 seems not convincing.
In this example, the only difference between task 1 and task 2 is the sign of the input. Since each task has a separate classifier, we can suppose a simple scenario where network has only one Linear Layer followed by a classifier. Since S2=S1, W2=W1 after learning task 2. Though W1x1 = -W2x2, we can adapt the classifier of task 2 which is task-specific to ensure Wc2 = -Wc1  and  make sure task 2 can be classified correctly
The example seems want to illustrate that the ideal output of task 2 should be the same as task 1, i.e., W1x1 = W2x2. But since the input has changed, why cannot the output change?
2.The way to measure the correlation between tasks.
Denote ΔX’∈R^(m×n), X∈R^(n×b), Δ∈R^(m×b) as the gradient spanned by the input batch of task t, where  is the input dimension and  is the label dimension. It’s projection onto the subspace of task j can be written as ΔX’ BjBj’. Perform SVD on X, the gradient projection can be rewritten as Δ(U∑V’)’ BjBj’, which is δU’BjBj’. Why use the projection of a vector spanned by U to measure the correlation instead of the base U?
3.As said in the paper, trust region aims to select strongly correlated tasks. But in Session ‘Impact of selected tasks in trust region’, it is obvious that trust region only selects most close tasks even if they are not strongly correlated, and this phenomenon can be seen in the task-wise experiment and higher layers in the layer-wise experiments, which makes me doubt the effectiveness of the trust region selection.
4.In the Session 4.2, this paper says that if task t is strongly correlated with task j, the weight projection on subspace of Sj is important for task t. This view seems like an assumption which is not proved. It would be more than welcome if more convincing proof could be provided.

**Summary Of The Paper:**

Some existing methods put restrictive constrains on the optimization space of the new task to prevent catastrophic forgetting, which may lead to unsatisfactory performance for the new tasks.
This paper aims to facilitate the forward knowledge transfer based on an efficient characterization of task correlation. Main contributions can be summarized as follows:
1.Introduce a novel notion of ‘trust region’ based on the norm of gradient projection onto the subspace spanned by task inputs to measure task correlation.
2.Proposed a novel approach for the new task to leverage the knowledge of the strongly correlated old tasks in the Trust Region through a scaled weight projection.
3.Developed a continual learning approach, trust region gradient projection(TRGP) based on the introduced Trust Region, scaled weight projection and a module to construct task input subspace.
4.Compared to related state-of-the-art approaches, TRGP achieves substantial performance improvement on all Benchmarks.

**Summary Of The Review:**

This paper put forward the main problem of existing methods and provides a detailed discussion. The author proposes a novel solution to this problem, which achieves substantial performance improvement on all benchmarks and overwhelms its baseline method. But some of the claims seem not convincing and some seem lack illustration.

---

> ### Author Response · Authors · 2021-11-22
> **Reply to Reviewer jo3a (2/2)**
>
> Q4: If task t is strongly correlated with task j, the weight projection on subspace of $S_j$ is important for task t.
>
> A4:  This view does have a solid theoretic foundation; for rigorous proofs, please refer to "Finite-dimensional vector spaces" (PR Halmos, 2017). In what follows, we outline the basic ideas behind this. Specifically, i) two strongly correlated tasks $i$ and $j$ usually share a lot of common features, which implies that their subspaces $S_i$ and $S_j$ share many common bases, denoted by $\{u_c\}$. As the weight projection on each subspace is the linear combination of the weight projection on bases, the weight projections for these two tasks, i.e., $proj_{S_i}(W)$ and $proj_{S_j}(W)$, share a lot of common components, i.e., $Wu_c u'_c$. Consequently, the weight projection change for one task will have more impact on the other task. ii) On the other hand, if the tasks are less correlated, fewer bases for these two tasks are same and more are orthogonal. Therefore, more components in the weight projections are orthogonal between these two tasks, and the weight projection change for one task will have less impact on the other task.

---

> > ### Comment · Reviewer_jo3a · 2021-12-06
> > **Thanks for the detailed explanation.**
> >
> > I have no further questions.

---

> ### Author Response · Authors · 2021-11-22
> **Reply to Reviewer jo3a (1/2)**
>
> Thank you for your thorough reviews and constructive comments. Per your suggestions, we have made the following major revision: (1) added new experiments on Split-MiniImageNet in Table 1 in section 5.2,  (2) added the standard deviation in Table 4 in Appendix B.2, (3) added the forward transfer comparison in Table 5-8 in Appendix B.3, (4) added training time comparison in Table 9 in Appendix B.4, (5) added the learning curves in Figure 9 and 10 in Appendix B.5, and (6) made various revisions throughout the paper based on all reviewers’ comments. All our changes are highlighted with blue-colored texts. New comments on these changes are very welcome!
>
>
> Q1: About the toy example 1.
>
> A1: Good observation! And you are right when each task has a separate classifier. However, if this is not the case, e.g., when two tasks share the same classifier as the single head network considered in GPM (Saha et al. 2021) for PMNIST and only differ in the signs of the inputs, the network model will not be updated for task 2 to address forgetting of task 1 with orthogonal projection, leading to suboptimal performance for task 2. We have clarified this in the revision.
>
> The purpose of the toy example 1 is to illustrate the cases where orthogonal projection may fail. This would motivate our conjecture that naive orthogonal projection could possibly compromise the learning performance of the new task that is strongly correlated with old tasks, which is also intuitive because restrictive constraints are put on the optimization space of the new task in orthogonal projection. Fortunately, a simple remedy for the toy example 1 is that we can use a scaling matrix to scale the network weights by -1 and reuse what have been learnt in task 1. This consequently motivates our design of selecting most related tasks and reusing their weights through a scaled weight projection.
>
>
> Q2: Why use the projection of a vector spanned by $U$ to measure the correlation instead of the base $U$?
>
> A2: We sincerely thank you for pointing out another option of measuring the task correlation. The main reason that we use the gradient projection instead of the bases is to reduce the computation complexity. Specifically, the gradient with respect to the input can be easily and quickly computed with one forward-backward pass; and in contrast, performing SVD on the input for every task to extract the bases will increase the computation complexity, especially when the trust region is layer-wise (need to perform SVD on the representations at each layer).
>
>
> Q3: Effectiveness of the trust region selection.
>
> A3: Thank you for the careful reading. We would like to clarify a few things about the trust region as follows.
>
> - As we mentioned in section 4.1 on page 5 just before the practical implementation, the layer-wise trust region could select different tasks for different layers and provide a more fine-resolution characterization of task correlations in terms of layer-level features, compared to the task-wise trust region. Therefore, in this work we consider the layer-wise trust region.
>
> - To better justify the benefit of the layer-wise trust region, we compare it with the task-wise trust region in section 5.3 "ablation study and analysis". It can be clearly seen that the layer-wise trust region outperforms the task-wise trust region. And as expected, the task-wise trust region cannot accurately select the most correlated tasks as the layer-wise counterpart does, as illustrated in Fig.7. Particularly, for Task 3 (not-MNIST), the layer-wise trust region selects Task 1 (MNIST) as Top-1 over Task 0 (CIFAR-10) and Task 2 (SVHN) in higher layers where the features are more specific compared to lower layers, and selects Task 1 (MNIST) as Top-2 in all layers. For Task 4 (Fashion MNIST), the layer-wise trust region selects Task 1 (MNIST) and Task 3 (not-MNIST) as Top-1 in almost all layers. It is also worth noting that because the trust region is selected in a one-shot manner based on stochastic gradients for computation efficiency, the task correlation is an estimate which could be noisy, and we believe there is more room for improvement about the task selection.

---

### Official Review · Reviewer_YFLo · 2021-11-03

**Correctness:** 4
**Technical Novelty And Significance:** 4
**Empirical Novelty And Significance:** 4
**Recommendation:** 8
**Confidence:** 4

**Main Review:**

**Strengths:**
The paper:
* addresses an important problem with a good solution
* is well motivated
* is theoretically sound
* is well written and easy to follow
* does a good job at concisely reviewing the important recent work on the topic

**Weaknesses:**
I don't see any major weaknesses in this paper. I have a few questions that I would like the authors to address (See below). Here are a few points that the paper can improve upon:

* providing the memory footprint of the algorithm
* comparing wall-clocks of TRGP/GP

**Questions for authors:** I appreciate it if the authors can clarify the following points and provide additional information.

1. The scaling matrix $Q_{j,t}^l$ plays a crucial role in your approach. This matrix is balancing the freezing/unfreezing process. Why wouldn't the network always choose $Q$s such that all parameters in the Trust Region are unfrozen? This would presumably lead to the least loss for the current task (as it provides the maximum capacity for the task). If so, why should one optimize $Q$ as opposed to simply fixing it to unfreeze all parameters in the trust region?

2. Could you please comment on the memory footprint of your algorithm? Also, could you please provide a head-to-head wall-clock comparison between GP and TRGP?

3. Last question is also about unfreezing and matrix $Q$. If my understanding is correct, all tasks in the Trust Region (for each layer)  are currently treated as equally important. For instance, if, for a certain layer, Task A and Task C have a correlation of 1.0 (same tasks), and Task B and Task C have a correlation of 0.8,  they are both added to the trust region of Task C, but the information that Task A is more important to unfreeze is missing and is left up to the optimization on $Q$ to decide this by itself.  Do you think there is a benefit in incorporating the correlation values into your Trust Region?


**Summary Of The Paper:**

**Summary:** The paper focuses on gradient projection (GP) for incremental learning. The authors motivate their work by stating that while approaches based on GP lead to superior performance in overcoming catastrophic forgetting, they suffer from a significant drawback. In GP, once one calculates the subspaces spanned by layerwise inputs for a task, say Task A, the network is then forced to only update the weights in an orthogonal direction to these subspaces for learning a subsequent task, say Task B (hence keeping important parameters for Task A intact and overcome catastrophic forgetting). However, suppose Task B and Task A are similar (an extreme case is when Task B is the continuation of Task A!). In that case, we know that the essential parameters for Task A are likely to be important for Task B and that the network could benefit by continuing to update the important parameters for Task A, which is not allowed in GP algorithms. This behavior has two consequences: 1) intransigence, i.e., the network won't be able to learn Task B as effectively as possible, and 2) the network will have reduced backward transfer. The backward transfer issue is apparent in the extreme case when Task B is a continuation of Task A, and the network's performance on Task A would have improved if it was able to learn Task B using the important weights for Task A!  The paper addresses this issue, with a simple, yet practical, solution. For a new task, the authors calculate the correlation between the subspaces calculated for old tasks and the new task (layerwise), keep track of the most correlated previous tasks for each layer, and denote them as (layerwise) Trust Regions. Next, the authors propose a scaled weight projection that allows for unfreezing the important parameters in the Trust Region of the new task, while learning this task, and learn the corresponding weights as part of the optimization process. Finally, the authors report results on Permuted MNIST (PMNIST), CIFAR100 Split, CIFAR100 Sup, and a sequence of 5-Datasets with 10-class classification which includes CIFAR-10, MNIST, SVHN, not-MNIST, and Fashion MNIST, in comparison with GP methods, regularization-based methods, and memory replay method. They show consistent improvement in accuracy, and more interestingly in backward transfer.


**Summary Of The Review:**

**Overall evaluation:**  I think the paper addresses a very important problem. Generally speaking, non-memory-replay-based methods for overcoming catastrophic forgetting could suffer from intransigence, which is the inability to learn new tasks due to increased stiffness/rigidity of the network. This reduces both forward and backward transfer, especially when the tasks are similar (e.g., revisiting an old task). This paper provides a rational solution to this problem, which provides consistent numerical improvement over the state-of-the-art. From an editorial point of view, the paper is well-written, easy to follow, and it provides a good overview of the recent literature on the topic. Hence, I think this is a good paper and vote for its acceptance.

---

> ### Author Response · Authors · 2021-11-22
> **Reply to Reviewer YFLo**
>
> Thank you for your thorough reviews and constructive comments. Per your suggestions, we have made the following major revision: (1) added new experiments on Split-MiniImageNet in Table 1 in section 5.2,  (2) added the standard deviation in Table 4 in Appendix B.2, (3) added the forward transfer comparison in Tables 5-8 in Appendix B.3, (4) added training time comparison in Table 9 in Appendix B.4, (5) added the learning curves in Figure 9 and 10 in Appendix B.5, and (6) made various revisions throughout the paper based on all reviewers’ comments. All our changes are highlighted with blue-colored texts. New comments on these changes are very welcome!
>
> Q1: Why should one optimize Q as opposed to simply fixing it to unfreeze all parameters in the trust region?
>
> A1: Many thanks for the insightful comments. In fact, our approach has gone  beyond choosing $Q$ to unfreeze all parameters, and it also  scales the weights in a favorable way for the new task by optimizing $Q$. Particularly, to find the appropriate scaling coefficients in $Q$ and then maximize the performance for the new task, a fixed $Q$ is not sufficient; instead, the matrix $Q$ should be optimized together with the network weights as shown in the optimization problem Eq. (8) (see section 4.4 in page 6).
>
>
> Q2: Comment on the memory footprint and provide a head-to-head wall-clock comparison between GPM and TRGP.
>
> A2: Per the reviewer's suggestion, we have added the computational complexity in Appendix B.4.
>
> - In terms of the memory, the major difference between TRGP and GPM is that TRGP requires additional memory to store the scaling matrices for each task. However, since the dimension of the scaling matrix is the same with the number of the extracted bases for the input subspace, which is usually small and controllable by the matrix approximation accuracy $\epsilon_{th}$ in Eq. (7), the memory increase is marginal and controllable. Particularly, the memory usage of TRGP can be further reduced by only learning the scaling matrices for the convolutional layers.
>
> - We compare the training time between TRGP and other baselines on relatively complex task sequences in Table 9 in Appendix B.4. For example, for CIFAR-100 Split, TRGP takes around 60\% more time than GPM, is comparable with HAT and ER\_Res, and takes less time than OWM and EWC; for 5-Datasets, TRGP takes around 20\% more time than GPM, but is much faster than other baselines including EWC, HAT, A-GEM and ER\_Res.
>
> Q3: Benefit in incorporating the correlation values into your trust region.
>
> A3: Thank you for pointing out this interesting direction. We have the following thoughts on this.
>
> Our approach selects tasks to the trust region in a one-shot manner using the stochastic gradient calculated with a batch of samples. Since the gradient is noisy, the task correlation obtained here is just an estimate, and it is infeasible to precisely evaluate the correlation ranking for tasks in the trust region. If the correlation values could be precisely quantified, incorporating these values should   help more. However, this interesting problem is non-trivial, and we will further investigate it in the future work based on your insightful comments.

---

> > ### Comment · Reviewer_YFLo · 2021-11-27
> > **Thank you for the rebuttal**
> >
> > First, I would like to thank the authors for providing answers to the raised comments/concerns. After reading all the reviews, and the authors' responses to all reviewers, I think this is a good paper and worthy of publication at ICLR.
> >
> > I have one minor follow-up question related to my Q3 and Reviewer jo3a's Q3 on the effectiveness of calculating task correlations. The question is regarding "when" the trust region is being calculated. Currently, the trust region is calculated once and at the onset of the new task. I am wondering about the need for dynamically tracking the trust region throughout the learning process. Let me elaborate below.
> >
> > Consider the PMNIST tasks. Two tasks in PMNIST are fully correlated if the first layer of the network solves the permutation problem. However, if we calculate the correlation (as proposed in the current work) at the onset of learning the second task, the tasks would potentially have layerwise correlations close to zero (not similar at all)! After a few optimization steps, I speculate that the earlier layers of the network could overcome the domain gap, and the later layers of the network would become highly correlated. Hence, it seems that calculating the trust region, not in a one-shot manner could be beneficial.
> >
> > This last comment is more of a curiosity for me, and I appreciate the authors' thoughts on it.

---

> > > ### Author Response · Authors · 2021-11-28
> > > **Reply to Reviewer YFLo**
> > >
> > > Thank you for your prompt response and insightful comments! We have the following thoughts on this interesting direction.
> > >
> > > - The underlying rationale for the trust region is as follows: Since the input subspaces for two strongly correlated tasks are highly correlated and the stochastic gradient descent updates lie in the input subspace (see e.g., Zhang et al. 2021, Saha et al. 2021), the layer-wise gradient of the new task should be highly correlated with the input subspace of the correlated 'old' (existing) tasks for the corresponding layer. For the noisy stochastic gradient, the computed correlation for strongly correlated tasks would be small with small possibility. To improve this part further, it would be better to evaluate the average gradient at each layer during a time window after a few optimization steps, as suggested insightfully by the reviewer.
> > >
> > > - As pointed out by the reviewer, it would also be beneficial to dynamically track the trust region throughout the learning process, which could lead to a more accurate selection of the most correlated tasks. We expect that this would pose some nontrivial challenges on learning the scaling matrices, because the scaling matrices need to vary accordingly with the selected old tasks.
> > >
> > > Overall, we sincerely thank the reviewer for pointing out this interesting direction for improving the trust region, and we believe that such a proposal with careful designs could further improve the performance of TRGP. Given that this interesting direction is non-trivial, we will further investigate it in the future work based on your insightful comments.

---

### Decision · Program_Chairs · 2022-01-20

**Decision:**

Accept (Spotlight)

**Comment:**

The submission addresses the problem of whether or not to update weights for a previous task in continual learning.  The approach is to specify a trust region based on task similarity and update weights only in the direction of the tasks that are similar enough to the current one.  The paper was on the balance well received (3/4 reviewers recommended acceptance, 2 with scores of 8) and complemented for its simple but effective approach, and good discussion of related literature.  The submission attracted a reasonable amount of engagement and discussion between reviewers and authors, which should be taken into account in the final version of the paper.